# Hydrogen Gas Inhalation Improved Intestinal Microbiota in Ulcerative Colitis: A Randomised Double-Blind Placebo-Controlled Trial

**DOI:** 10.3390/biomedicines13081799

**Published:** 2025-07-23

**Authors:** Takafumi Maruyama, Dai Ishikawa, Rina Kurokawa, Hiroaki Masuoka, Kei Nomura, Mayuko Haraikawa, Masayuki Orikasa, Rina Odakura, Masao Koma, Masashi Omori, Hirotaka Ishino, Kentaro Ito, Tomoyoshi Shibuya, Wataru Suda, Akihito Nagahara

**Affiliations:** 1Department of Gastroenterology, Juntendo University Faculty of Medicine, 2-1-1 Hongo, Bunkyo-ku, Tokyo 113-8421, Japan; t-maruyama@juntendo.ac.jp (T.M.); ke-nomura@juntendo.ac.jp (K.N.); m.haraikawa.bk@juntendo.ac.jp (M.H.); m.orikasa.tr@juntendo.ac.jp (M.O.); r.odakura.sp@juntendo.ac.jp (R.O.); m-koma@juntendo.ac.jp (M.K.); ma-omori@juntendo.ac.jp (M.O.); h.ishino.jl@juntendo.ac.jp (H.I.); k.ito.wj@juntendo.ac.jp (K.I.); tomoyosi@juntendo.ac.jp (T.S.); nagahara@juntendo.ac.jp (A.N.); 2Innovative Microbiome Therapy Research Center, Juntendo University Graduate School of Medicine, 2-1-1 Hongo, Bunkyo-ku, Tokyo 113-8421, Japan; 3Laboratory for Symbiotic Microbiome Sciences, RIKEN Centre for Integrative Medical Sciences, Yokohama 230-0045, Japan; rina.kurokawa@riken.jp (R.K.); hiroaki.masuoka@riken.jp (H.M.); wataru.suda@riken.jp (W.S.)

**Keywords:** hydrogen gas inhalation, ulcerative colitis, intestinal microbiota

## Abstract

**Background/Objective**: Dysbiosis is implicated in the pathogenesis of ulcerative colitis. Hydrogen has been reported to promote intestinal microbiota diversity and suppress ulcerative colitis progression in mice models. In this study, we investigated changes in the intestinal microbiota, therapeutic effects, and safety of hydrogen inhalation in patients with ulcerative colitis. **Methods**: In this randomised, double-blind, placebo-controlled trial, 10 active patients with ulcerative colitis (aged ≥20 years; Lichtiger’s clinical activity index, 3–10; and Mayo endoscopic subscores ≥1) participated, and they were assigned to either a hydrogen or air inhalation group (hydrogen and placebo groups, respectively). All patients inhaled gas for 4 h every day for 8 weeks. Subsequently, we performed clinical indices and microbiota analyses using the metagenomic sequencing of stool samples before and after inhalation. **Results**: There was significant difference in the sum of the Mayo endoscopic subscores before and after inhalation in the clinical assessment indices. The hydrogen group showed higher α-diversity (*p* = 0.19), and the variation in β-diversity was markedly different, compared to the placebo group, in intestinal microbiota analysis (*p* = 0.02). Functional gene analysis revealed 115 significant genetic changes in the hydrogen group following treatment. No inhalation-related adverse events were observed. **Conclusions**: Hydrogen inhalation appeared to improve intestinal microbiota diversity; however, no clear therapeutic effect on ulcerative colitis was observed. Further studies are needed, and hydrogen inhalation may possibly lead to a logical solution combined with microbiome therapy, such as faecal microbiota transplantation, with fewer adverse events.

## 1. Introduction

The relationship between ulcerative colitis (UC) and factors such as immunity, environment, genetics, and diet has been recognised [1]. Patients with UC experience a reduction in the diversity and richness of their intestinal microbiota, leading to a microbiome imbalance known as dysbiosis [2,3].

Recently, the prevalence of UC has been rising globally, with a notable increase in Asia and South America, whereas it remains high in Western countries [4]. Several biological and molecular target drugs have been used to treat UC; however, they have various adverse effects, including infection risk associated with immunosuppression and thrombosis [5]. Treatments with fewer side effects are preferred for UC considering the increase in patients with elderly-onset UC [6].

Microbiota-based therapies, such as faecal microbiota transplantation (FMT), have gained attention in the treatment of UC, with numerous reported positive outcomes [7]. FMT is a minimally invasive treatment that restores normal intestinal microbial function by transplanting gut microbiota from healthy donors in patients with diseases of gut microbiome dysbiosis [8]. Nevertheless, the long-term survival of the gut microbiota restored by FMT remains a challenge, underscoring the importance of strategies to maintain gut microbiota post FMT [9,10].

Hydrogen (H_2_) exhibits antioxidant and anti-inflammatory properties and has been reported to be effective against various diseases in mice, including hypertension, acute respiratory distress syndrome, small intestine transplantation, and colitis. In human clinical practice, its efficacy in targeting coronavirus disease and post-cardiac arrest care has been demonstrated [11,12,13,14,15,16].

Furthermore, it has been suggested that H_2_ may be effective in managing diseases involving intestinal microbiota. H_2_ ingestion alters the intestinal microbiota diversity, affecting food and water intake in mice [17]. Similarly, reports show that drinking H_2_-dissolved alkaline electrolysed water in humans causes changes in the intestinal microbiota and improves normal stool characteristics [18]. *Firmicutes* and *Bacteroidetes* are anaerobic bacteria abundant in the colon that produce H_2_ as a by-product of carbohydrate degradation. H_2_ provides a substrate for hydrogenotroph growth, thereby increasing their abundance and promoting H_2_ metabolism in the intestine, which alters intestinal microbiota [17].

Previously, a study reported that H_2_ contributed to the gut’s metabolite milieu, while it entered circulation through the gut mucosa wall and was transported to various organs [13]. The area under the curve of H_2_ when administered in gaseous form is significantly higher than that of other forms of administration, such as H_2_-rich water [19,20].

A previous study revealed that H_2_ could reduce oxidation-dependent tissue damage in the small intestine caused by ischaemic reperfusion following small intestine transplantation and it inhibited the overgrowth of *Enterobacteriaceae* [21,22]. H_2_ alleviated clinical and pathological inflammation in the UC mouse model induced by dextran sodium sulphate (DSS) [23,24,25]. However, the impact of H_2_ on UC in humans has not been clarified.

In this randomised controlled trial, we aimed to investigate the therapeutic effects of H_2_ inhalation and its impact on the intestinal microbiota.

## 2. Materials and Methods

### 2.1. Study Design

We conducted a placebo-controlled, randomised, double-blind, parallel-group (1:1) clinical trial on H_2_ gas inhalation for the treatment of UC.

Patients were recruited using posters in our examination rooms and on our department homepage on the hospital’s website. The participants voluntarily decided to participate. This study was registered in the UMIN Clinical Trial Registry (UMIN-CTR; ID: UMIN000042017; registered on 5 October 2020). The study was approved by the local ethics committee of Juntendo University School of Medicine on 13 August 2020 (institutional review board no. 20-079). Informed written consent was secured from all participants prior to their involvement in the study.

We diagnosed UC based on established clinical, endoscopic, and histological criteria. Participants eligible for inclusion were individuals aged 20 years or older, of either sex, who had previously undergone endoscopic evaluation, had a confirmed diagnosis of active UC, and a Lichtiger’s clinical activity index (CAI) of 3–10 and Mayo endoscopic subscore (MES) ≥1. UC medications remained unchanged for 8 weeks prior to the baseline assessment. We excluded intestinal superinfections in all patients using blood tests. Additional exclusion criteria comprised malignancies, the presence of severe health conditions, pregnancy, and involvement in other clinical trials.

### 2.2. Enrolment

We registered 10 Japanese participants with UC from 13 November 2020 to 16 August 2021. Table 1 indicates the characteristics of the participants at baseline.

This was an exploratory study was conducted in the absence of sufficient prior data; therefore, the initial target size was set as 20 participants within the designated study period. To examine the safety and efficacy of the study intervention, an interim analysis was conducted after enrolling the first 10 participants. As the clinical evaluation did not show clear therapeutic effects, continuing the interventional trial without evident clinical improvement could not be justified from both scientific and ethical standpoints. Consequently, this study was terminated after enrolling 10 participants.

### 2.3. Randomisation and Blinding

The participants were sequentially assigned by T.K., a member of MiZ Co. that provided both the placebo and H_2_-producing machine. Apart from assigning participants, T.K. was not involved in this study. The participants and authors who assessed the results were blinded until all participants had completed their participation in the study.

### 2.4. Procedures

We assigned half of the participants to a H_2_ gas inhalation group (H_2_ group) and the other half to the placebo-control group (placebo group). The Hydrogen Gas Inhaler Jobs-α from MiZ Co. (Kamakura, Kanagawa, Japan) was used as the H_2_-producing machine, generating 5–6% H_2_ at 4 L/min via electrolysis. For the placebo group, a visually identical device was used, but its electrolysis electrodes were removed, producing only air at the same flow rate of 4 L/min. All participants inhaled gas for 4 h daily for 8 weeks. This corresponded to a daily hydrogen intake of approximately 48–57.6 L of hydrogen, inhaled at the participants’ homes, amounting to a cumulative dose 2688–3225.6 L over the 8-week period. We checked the gas inhalation time when the participants returned to the hospital after 8 weeks.

As the primary endpoints, we evaluated the changes in the Mayo score, CAI, MES, and sum of the MES from baseline to after 8 weeks, while monitoring for adverse events, to compare the outcomes of the study interventions. Endoscopic mucosal evaluation was performed before and after 8 weeks. Remission was defined when their Mayo score improved to ≤2 after 8 weeks.

As the secondary endpoints, we evaluated the changes in histological indices and blood test results. In addition, microbial analysis was conducted at baseline and after 8 weeks with faecal samples collected from the participants. Specifically, we assessed α-diversity and β-diversity to evaluate the changes in the gut microbiota induced by hydrogen inhalation. α-diversity is an index that assesses the microbial diversity within a single sample, reflecting the richness and evenness of microbial species while evaluating the ecological complexity of the microbiota. β-diversity measures the differences in microbiota composition between different samples, providing insights into intergroup variation in microbial community structure.

We assessed blood test results (red blood cell, white blood cell, and platelet counts, leukocyte fraction, haemoglobin, erythrocyte sedimentation rate, total protein, albumin, alkaline phosphatase, total bilirubin, aspartate and alanine transaminases, serum urea nitrogen, lactate dehydrogenase, creatinine, chloride, potassium, sodium, uric acid, choline esterase, and C-reactive protein) at baseline and after 8 weeks (Figure 1).

### 2.5. Statistical Analysis

The treatment score for each group represented the between-group differences. We used the Wilcoxon signed-rank sum test or Mann–Whitney U test for statistical analyses. Data analyses were performed with SPSS version 19 (IBM Corp., Armonk, NY, USA). Statistical significance was set at *p* < 0.05.

We performed microbial analysis using Library Preparation for whole genome shotgun sequencing and metagenomic analysis at the Laboratory for Symbiotic Microbiome Sciences, RIKEN Centre for Integrative Medical Sciences.

The whole-genome shotgun sequencing and metagenomic analysis of faecal samples were carried out by first extracting stool DNA, followed by the library and sequencing of the extracted DNA for next-generation sequencing. The sequencing data were then pre-processed and assembled to reconstruct the genetic information of the microorganisms and identify their classification and function (Appendix A).

## 3. Results

Ten patients were enrolled in this study, and at baseline, there were no significant differences in age, sex, disease location, CAI, Mayo score, or blood test results between the H_2_ and placebo groups (Table 2).

All patients completed this trial (five patients each in the H_2_ and placebo groups, respectively) without any problems, including no malfunctioning of the gas inhaler.

There were no significant differences in the inhalation time (215.8 ± 120.9 and 235.4 ± 72.5 h, respectively, in the placebo and H_2_ groups).

### 3.1. Primary Endpoints

The remission induction rate in the H_2_ group was 3/5 cases, whereas that in the placebo group was 1/5 cases (*p* = 0.24). The Mayo score decreased from 4.4 ± 1.6 to 3.0 ± 1.3 in the H_2_ group. The Mayo score changed from 5.0 ± 1.4 to 3.8 ± 1.5 in the placebo group. Similarly, the CAI decreased from 5.2 ± 2.3 to 3.2 ± 2.4 in the H_2_ group and from 5.2 ± 1.2 to 3.6 ± 1.2 in the placebo group. There were no significant differences in the Mayo score or CAI between the two groups at 8 weeks (*p* = 0.80). The between-group difference in the change in the sum of the MES was statistically significant (*p* = 0.02). The effect size (Cohen’s d) was calculated as 1.73, with a 95% confidence interval of [0.15, 2.65], indicating a potentially moderate to large effect. However, the wide confidence interval reflects the small sample size and associated uncertainty (Appendix A).

### 3.2. Secondary Endpoints

There were no significant differences in the blood test results or histological indices before and after inhalation (Appendix A).

### 3.3. Adverse Events

No adverse events were observed during or after H_2_ inhalation.

### 3.4. Microbial Analysis

The proportion of intestinal microbiota was compared between baseline and after inhalation (Figure 2). Visually, the placebo group showed no changes, whereas the H_2_ group experienced a change in the relative abundance of microbiota. Changes in intestinal microbiota were not observed in the placebo group, whereas in the H_2_ group, *Erysipelotrichaceae* genus *incertae sedis* and *Erysipelotrichaceae* species *incertae sedis* (*p* = 0.03 and 0.02, respectively) were increased, and *Atopobium parvulum* and *Erysipelatoclostridium ramosum* (*p* = 0.03 and 0.03, respectively) were decreased (Figure 3).

Microbial analysis revealed that the Shannon index, an indicator of α-diversity and species richness, exhibited greater diversity after H_2_ inhalation (*p* = 0.19); however, the change was not statistically significant (Figure 4a). There was a significant variation in β-diversity in the H_2_ group compared to the placebo group (*p* = 0.02) (Figure 4b,c).

Functional gene analysis based on the Kyoto Encyclopedia of Genes and Genomes (KEGG) showed 115 significant genetic changes in KEGG orthology (KO) between baseline and 8 weeks after inhalation in the H_2_ group. A total of 83 of 115 KOs were significantly enriched in the H_2_ group, whereas 32 KOs were depleted. In the placebo group, 40 of 99 KOs were significantly enriched, whereas 59 KOs were depleted. Two KOs (K24217 and K00849) showed opposite variables to those in the placebo and H_2_ groups (Figure 5).

## 4. Discussion

To the best of our knowledge, this is the first study to explore the benefits of H_2_ inhalation in humans with UC. The Mayo score showed a non-significant trend toward improvement in the H_2_ group; however, the clinical outcomes did not differ significantly between the two groups. Moreover, our study revealed some trends in microbial composition, such as an improvement in beta diversity, but these findings should be interpreted with caution due to the lack of statistical significance in alpha diversity and the small sample size. Collectively, these observations highlight the exploratory nature of this study and suggest directions for future research.

Previous studies have reported that H_2_ inhalation improves symptoms of DSS-induced colitis in a mice model of UC [23]. There is a marked difference in the composition of the intestinal microbiota between mice and humans, as well as in the concentration of H_2_ inhalation as mice inhale H_2_ in higher concentrations than humans do. Therefore, this difference may not have led to a clear improvement in symptoms in humans [26]. Previous treatments with 5-amino salicylic acid may have influenced microbiota variability, limiting the interpretability of microbial shifts. Similarly, changes in intestinal microbiota before and after treatment with corticosteroids and anti-tumour necrosis factor-α inhibitors have been reported. Therefore, it is likely that the composition of the intestinal microbiota had high variability [27,28].

The high therapeutic efficacy and safety of FMT for UC are known. Therefore, there is a pressing need to develop interventions for maintaining the microbiota diversity recovered by FMT. Currently, a growing body of research is exploring the potential of probiotics and prebiotics in the treatment of UC. However, the evidence so far has not demonstrated any significant improvement in diversity [29]. Recent studies have shown that a reduction in gut microbiota diversity is associated with UC severity. FMT introduces healthy donor microbiota into patients and can help restore microbial diversity, with greater increases in diversity following FMT being associated with better clinical outcomes [30]. In this study, although hydrogen gas inhalation did not lead to significant changes in the intestinal environment or clinical improvements, it successfully increased β-diversity variation in the gut microbiota compared to the placebo group. In addition, the number of KOs enriched after intervention in the H_2_ group was higher than the number of KOs depleted. In contrast, the number of KOs enriched after intervention in the placebo group was less than the number of KOs depleted. While KEGG pathway enrichment was not clearly observed, several individual KOs showed changes following H_2_ inhalation, including K00242, K18120, K00135, and K14534. These KOs are involved in butanoate metabolism and promote intestinal epithelial repair and barrier function, regulate immune responses, and influence the composition of the gut microbiota. While these KO-level changes do not form coherent pathways, they may reflect modest functional shifts in the microbiome associated with H_2_ exposure. This modest change may represent a small but meaningful step toward achieving the therapeutic potential of FMT in UC. Combining hydrogen gas inhalation with FMT may enhance therapeutic efficacy and warrants further investigation. There is a future possibility that FMT combined with H_2_ inhalation, which maintains the intestinal microbiota, could be a new treatment method that does not require immunosuppression. To strengthen the clinical relevance of hydrogen gas inhalation therapy, future trials should consider longer-term follow-up to assess the durability of microbial shifts and clinical responses. In addition, combining hydrogen gas therapy with microbiota-targeted interventions such as FMT or probiotics may provide synergistic effects. Stratification by UC subtypes or disease activity may also help to clarify which patient populations are most likely to benefit from such therapies.

This study has several limitations. First, the sample size was small, as the trial was terminated after the interim analysis of 10 participants showed no clear clinical improvement. Although microbiota-related changes were observed, the limited sample size provided reduced statistical power. Second, all patients continued standard therapies during the intervention, which may have confounded the results. Discontinuing these treatments to isolate H_2_ effects was deemed ethically inappropriate. Thus, H_2_ inhalation was administered as an adjunctive therapy. Third, the lack of long-term follow-up limits the assessment of sustained efficacy and safety. In some cases, additional treatments were introduced, preventing consistent long-term observation. Finally, the observed MES reduction should be interpreted cautiously due to the small sample size and potential Type I error. Larger, adequately powered studies are necessary to confirm these preliminary findings.

In this study, no adverse events were documented, and an increase in the variation in intestinal microbiota diversity was observed. Nonetheless, there is a possibility that H_2_ inhalation combined with microbiome therapy such as FMT may lead to a rational solution for UC management with fewer adverse events in the future.

## Figures and Tables

**Figure 1 biomedicines-13-01799-f001:**
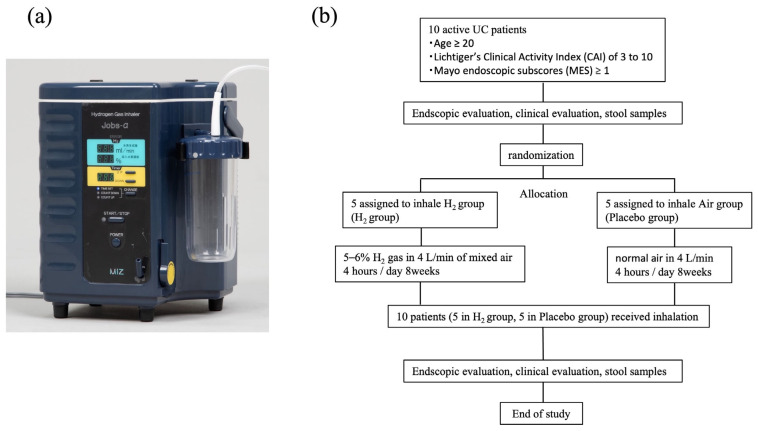
Hydrogen-producing machine and patient flow diagram. (**a**) The hydrogen gas inhaler Jobs-α from MiZ Co., Ltd., Kamakura, Japan, was used in this study. Jobs-α was generates 5–6% H_2_ gas at 4 L/min, mixed with air via electrolysis. The placebo machine is visually identical. The figure shows the display of the hydrogen delivery device. The upper numerical value indicates the hydrogen generation rate in milliliters per minute (mL/min), while the lower value shows the concentration of hydrogen in the inhaled gas (%). (**b**) Patient flow diagram. All patients inhaled H_2_ gas for 8 weeks. Endoscopic evaluation, clinical evaluation, and microbial analysis were performed at baseline and 8 weeks. UC, ulcerative colitis.

**Figure 2 biomedicines-13-01799-f002:**
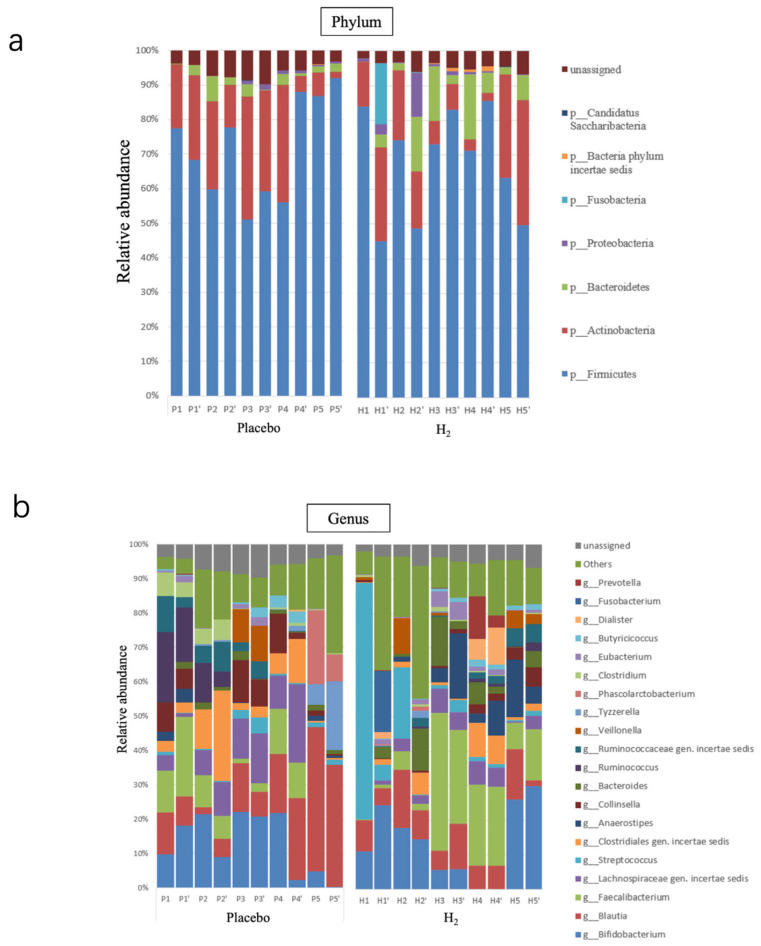
(**a**) The bar graph shows the relative abundance of intestinal microbiota at the phylum level in the placebo group (P1–P5) and hydrogen group (H1–H5) before and after inhalation (with prime, e.g., P1’, H1’). Each colour represents a different phylum. (**b**) The bar graph shows the relative abundance of the intestinal microbiota at the genus level in the same groups. Each colour represents a different genus. The hydrogen group shows minimal changes in microbiota composition after inhalation.

**Figure 3 biomedicines-13-01799-f003:**
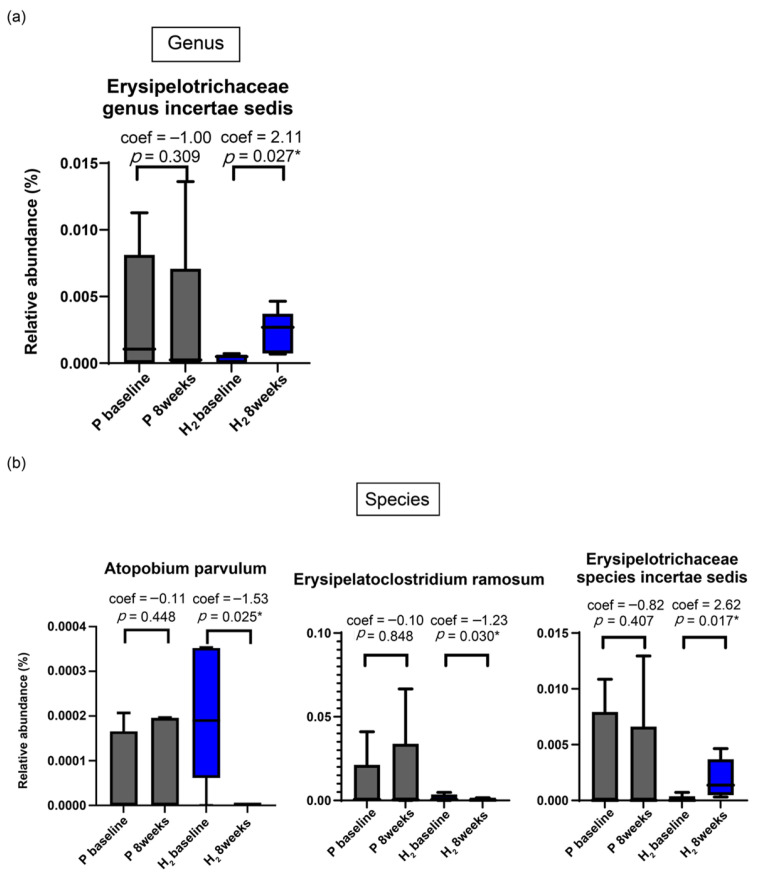
(**a**) Genus-level analysis shows an increase in *Erysipelotrichaceae* genus *incertae sedis* (MaAsLin2, *p* = 0.03). (**b**) Species-level analysis reveals a decrease in *Atopobium parvulum* and *Erysipelatoclostridium ramosum* (both *p* = 0.03) and an increase in Erysipelotrichaceae species *Incertae sedis* (MaAsLin2, *p* = 0.02). H_2_, hydrogen. * indicates *p* < 0.05, considered statistically significant.

**Figure 4 biomedicines-13-01799-f004:**
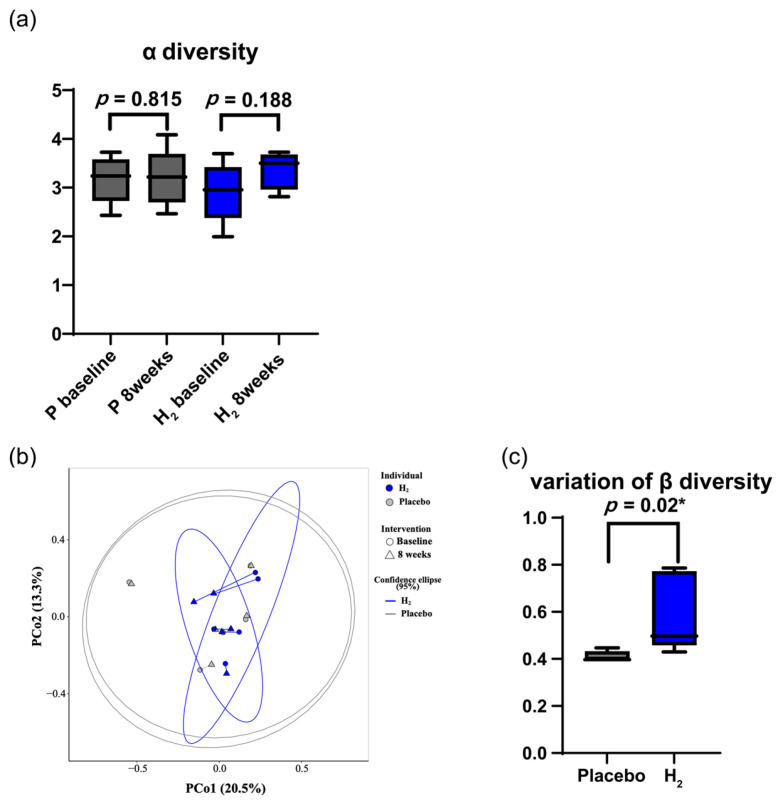
(**a**) The α-diversity of the intestinal microbiota was analysed. Although no significant difference is observed, the Shannon index shows increased diversity after H_2_ inhalation (the Wilcoxon signed-rank test, *p* = 0.19). (**b**) Principal coordinates analysis (PCoA) based on Bray–Curtis dissimilarity was performed to visualise β-diversity. PC1 and PC2 explain 20.5% and 13.3% of the variance, respectively. Each arrow represents the temporal shift of an individual subject from baseline (circle) to 8 weeks (triangle), with colour indicating group (blue: H_2_, grey: placebo). Due to the small sample size (n = 5 per group), confidence ellipses were not included, as they may not be statistically meaningful under these conditions. (**c**) Variation in β-diversity is shown. Compared to the placebo group, the hydrogen group exhibits a significant change in β-diversity (Wilcoxon rank sum test, *p* = 0.02). * indicates *p* < 0.05, considered statistically significant.

**Figure 5 biomedicines-13-01799-f005:**
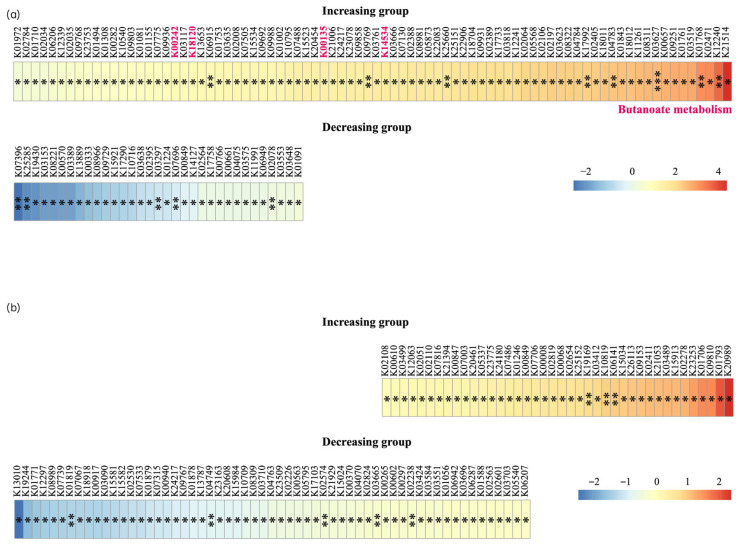
(**a**) In the hydrogen group, 115 significant KO changes are detected between baseline and 8 weeks (*p* < 0.05). Of these, 83 KOs are enriched (red) and 32 are depleted (blue). (**b**) In the placebo group, 99 significant KO changes are identified (*p* < 0.05); 40 are enriched (red) and 59 KOs are depleted (blue). KO, Kyoto Encyclopedia of Genes and Genomes Orthology. Statistical significance is indicated by asterisks: * *p* < 0.05, ** *p* < 0.01, and *** *p* < 0.001.

**Table 1 biomedicines-13-01799-t001:** The baseline characteristics of the study participants.

	Age (Years)	Sex	Duration of UC (Years)	Type of UC	Ongoing Treatments
H1	37	F	5	Proctitis	5-ASA, Topical corticosteroids
H2	39	F	4	Proctitis	5-ASA, Anti-TNF
H3	32	F	2	Extensive	5-ASA, Topical corticosteroids
H4	49	F	3	Left-sided	5-ASA, Topical corticosteroids
H5	40	M	8	Proctitis	5-ASA, Topical corticosteroids
Mean ± SD	39.4 ± 5.5		4.8 ± 1.9		
P1	38	F	6	Proctitis	5-ASA
P2	26	F	2	Extensive	5-ASA, Vedolizumab
P3	25	M	9	Extensive	Immunosuppressants
P4	56	M	35	Extensive	5-ASA, Topical corticosteroids
P5	39	M	18	Proctitis	5-ASA
Mean ± SD	36.8 ± 11.2		14 ± 11.7		
*p* value	0.69	0.52	0.19		

UC, ulcerative colitis; 5-ASA, 5-aminosalicylic acid; SD, standard deviation; H1–H5, participant IDs assigned to the hydrogen inhalation group; P1–P5, participant IDs assigned to the placebo group.

**Table 2 biomedicines-13-01799-t002:** The scores of blood test, endoscopic, and symptom scores at baseline for each group.

	Placebo Group at Baseline n = 5	Hydrogen Group at Baseline n = 5	*p*
WBC	7500 ± 1881.5	8480 ± 2391.2	0.54
Hb	14.8 ± 1.8	12.5 ± 0.9	0.06
Plt	284.2± 32.1	299.2 ± 54.2	0.65
CRP	0.12 ± 0.08	0.12 ± 0.1	0.93
Alb	4.6 ± 0.5	4.0 ± 0.3	0.14
Mayo score	5.0 ± 1.4	4.4 ± 1.6	0.59
MES	1.4 ± 0.5	1.0 ± 0	0.18
Sum of MES	2.4 ± 1.0	2.0 ± 0.6	0.53
CAI	5.2 ± 1.2	5.2 ± 2.3	1.0
UCEIS	2.8 ± 1.5	2.8 ± 1.0	1.0
Inhalation Times	215.8 ± 120.9	235.4 ± 72.5	0.79

WBC, white blood cell; Hb, haemoglobin; Plt, platelet; CRP, C-reactive protein; Alb, albumin; MES, Mayo endoscopic subscore; CAI, clinical activity index; UCEIS, ulcerative colitis endoscopic index of severity.

## Data Availability

The original contributions presented in this study are included in the article. Further inquiries can be directed to the corresponding author.

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
