# Peer review of "Hydrogen Gas Inhalation Improved Intestinal Microbiota in Ulcerative Colitis: A Randomised Double-Blind Placebo-Controlled Trial"

_biomedicines, 2025, doi:10.3390/biomedicines13081799_

Round 1

Reviewer 1 Report

Comments and Suggestions for Authors

This is an interesting research study. Current treatments on ulcerative colitis show obvious side effects. Authors tested the effects of hydrogen on patients with the ulcerative colitis, finding that hydrogen can significantly improve intestinal microbiota diversity. This study may offer a novel therapeutic strategy for treatment of ulcerative colitis due to no adverse effects. Overall, this study was written and organized well. However, I still have some comments as below before it can be accepted for publication.

Major comments

  1. In the method part, the hydrogen treatment was given to patients for 8 weeks. What is the main reason of 8 weeks window rather than other durations such as shorter or longer than 8 weeks? If hydrogen is really useful in future clinical treatment, the long treatment duration up to 8 weeks seem to be a disadvantage of broadening its application. So, I am wondering if shorter treatment timing will also cause similar results.
  2. It is not so clear about the “enrolment” part. Line 110, its claimed that 20 participants were the set number. However, in Table 1, only 10 patients were reported. This discrepancy is confusing for me. Also, if this study actually employed 10 participants for this report, such small number seems not sufficient to conclude compelling results.
  3. In the “procedures” part, authors showed that they used 5-6% H2 to treat patients to check its effects on ulcerative colitis. It is not clear the unit is in volume or percentage. This concentration is important for this study because it should indicate what concentration of hydrogen caused the reported microbiota diversity results as a clinical related study. Another question is about how authors give hydrogen to these patients. It should be made clear they are given by different machines or the same machines.
  4. In the figure 4, results of both α-diversity and β-diversity were specifically reported. However, what are α-and β-diversity? Authors should give more explanation in the text or in the method parts. How important are these parameters in reflecting the test results?

Minor comments

  1. Line 23, ofhydrogen to of hydrogen.
  2. For table 1, what does H1 to H5 mean? What does P1 to P5 mean? It is confusing to understand without careful reading of manuscript. It should be annotated below the table.
  3. Line 199, Figure S1b-e to Figures S1b-e. Same for line 227 Figure.
  4. All p value for statistic significance should be indicated as italic p. Figures use uppercase P while lowercase p is used elsewhere.

Author Response

Reviewer 1:

Major comments

1, In the method part, the hydrogen treatment was given to patients for 8 weeks. What is the main reason of 8 weeks window rather than other durations such as shorter or longer than 8 weeks? If hydrogen is really useful in future clinical treatment, the long treatment duration up to 8 weeks seem to be a disadvantage of broadening its application. So, I am wondering if shorter treatment timing will also cause similar results.

Response:

Thank you very much for your valuable question and suggestion. Many clinical studies on biologics (J Clin Med. 2025 Mar 25;14(7):2232) and FMT (Front Med. 2023 Jan 12:9:1049849) evaluating short-term treatment efficacy in ulcerative colitis also commonly adopt an 8-week observation window, which supported our decision. Additionally, we considered a longer intervention period for this study; however, given the inflammatory nature of ulcerative colitis and the treatment duration that patients can reasonably tolerate, we set the intervention period at 8 weeks.

Although it is possible that a shorter treatment duration might also be effective, our study did not observe clear benefits even with 8 weeks of hydrogen intervention. Therefore, we believe that it would be difficult to achieve meaningful therapeutic effects with a shorter duration.

2, It is not so clear about the “enrolment” part. Line 110, its claimed that 20 participants were the set number. However, in Table 1, only 10 patients were reported. This discrepancy is confusing for me. Also, if this study actually employed 10 participants for this report, such small number seems not sufficient to conclude compelling results.

Response:

We apologize for the insufficient explanation. As stated in the manuscript, we initially planned to recruit 20 participants. However, an interim analysis was conducted after enrolling 10 participants, and the results showed no clear clinical improvement. Therefore, we decided to terminate further recruitment at that point. In response to your suggestion, we have added a more detailed explanation of this in the Materials and Methods section, and the revised text has been highlighted in blue.

As you rightly pointed out, this study indeed has a small sample size. However, since the participants were patients with active disease, we considered it ethically and practically difficult to continue an intervention trial in the absence of observable clinical improvement.

              In response to your suggestion, we have added a detailed description of this limitation, which is highlighted in blue. The previous brief mention of this limitation has been removed to avoid redundancy in the last paragraph.

The revised content is as follows:

As the clinical evaluation did not show clear therapeutic effects, continuing the interventional trial without evident clinical improvement could not be justified from both scientific and ethical standpoints. Consequently, the study was terminated after enrolling 10 participants (Lines 115-118).

This study has several limitations. First, the sample size was small, as the trial was terminated after the interim analysis of 10 participants showed no clear clinical improvement. Although microbiota-related changes were observed, the limited sample size provided a reduced statistical power (Lines 315-318).

3, In the “procedures” part, authors showed that they used 5-6% H2 to treat patients to check its effects on ulcerative colitis. It is not clear the unit is in volume or percentage. This concentration is important for this study because it should indicate what concentration of hydrogen caused the reported microbiota diversity results as a clinical related study. Another question is about how authors give hydrogen to these patients. It should be made clear they are given by different machines or the same machines.

Response:

Regarding the procedure, we provided the same model of the hydrogen-generating device or placebo device to each participant (through postal service), who then performed inhalation at home. As described in the original manuscript, “Jobs-α was developed to generate 5–6% Hâ‚‚ in 4 L/min mixed air by electrolysis. All participants inhaled gas for 4 hours daily for 8 weeks.”  Therefore, the participants inhaled approximately 48–57.6 liters of hydrogen per day, totaling 2,688–3,225.6 liters over the 8-week period. We have added this clarification to the Materials and Methods  section.

The revised content is as follows:

This corresponded to a daily hydrogen intake of approximately 48–57.6 L of hydrogen, inhaled at the participants’ homes, amounting to a cumulative dose 2,688–3,225.6 L over the 8-week period (Lines 137-139).

4, In the figure 4, results of both α-diversity and β-diversity were specifically reported. However, what are α-and β-diversity? Authors should give more explanation in the text or in the method parts. How important are these parameters in reflecting the test results?

Response:

α-diversity is an index used to evaluate the microbial diversity within a single sample. It reflects the richness and evenness of microbial species, thereby assessing the ecological complexity of the microbiota. β-diversity measures the differences in microbial community composition between samples, helping to elucidate structural variations in the microbiota across different groups.

In our study, there was no significant differences in α-diversity before and after the intervention in either the placebo or hydrogen group. However, the β-diversity analysis demonstrated that the microbial changes in the hydrogen group were more pronounced compared to the placebo group. Since patients with ulcerative colitis generally exhibit significantly reduced gut microbial diversity, the observed change in β-diversity suggests that hydrogen inhalation may help improve the intestinal environment.

As per your suggestion, we have added these explanations in the Materials and Methods section. Please kindly review.

The revised content is as follows:

Specifically, we assessed α-diversity and β-diversity to evaluate the changes in the gut microbiota induced by hydrogen inhalation. α-diversity is an index that assesses the microbial diversity within a single sample, reflecting the richness and evenness of microbial species while evaluating the ecological complexity of the microbiota. β-diversity measures the differences in microbiota composition between different samples, providing insights into intergroup variation in microbial community structure (Lines 148-153).

Minor comments

1, Line 23, ofhydrogen to of hydrogen.

Response:

Thank you for your careful reading. We have confirmed that there is indeed a space between "of" and "hydrogen" in the original file. It may have appeared as a single word due to font rendering or formatting issues in the PDF. We will ensure this is clearly visible in the final version.

2, For table 1, what does H1 to H5 mean? What does P1 to P5 mean? It is confusing to understand without careful reading of manuscript. It should be annotated below the table.

Response:

H1–H5 represent the participant IDs assigned to the hydrogen inhalation group, and P1–P5 represent the participant IDs assigned to the placebo group. We have added this explanation below Table 1.

The revised content is as follows:

H1–H5, participant IDs assigned to the hydrogen inhalation group; P1–P5, participant IDs assigned to the placebo group (Lines 121,122).

3, Line 199, Figure S1b-e to Figures S1b-e. Same for line 227 Figure.

Response:

Thank you for your thorough review and suggestions. We have made the revisions, indicated in blue font, accordingly (Line 213, Line 240).

4, All p value for statistic significance should be indicated as italic p. Figures use uppercase P while lowercase p is used elsewhere.

Response:

Thank you for your suggestion. We have unified the notation of p throughout the manuscript. Kindly check.

Reviewer 2 Report

Comments and Suggestions for Authors

The manuscript presents results of a trial investigating the effects of hydrogen gas inhalation on intestinal microbiota and clinical outcomes in patients with ulcerative colitis. The study design is relevant, and the focus on microbiota modulation as a potential therapeutic target coincides with the latest trends in biomedicine.  The use of hydrogen gas as a therapeutic intervention in UC seems to be the new approach, and this study addresses an important gap in current treatment strategies. The study is randomized, double-blind, and placebo-controlled, which strengthens the validity of the findings. The inclusion of metagenomic sequencing and functional analysis provides valuable insights into the mechanisms underlying the observed effects. The results are presented in a clear and accessible manner.

There are still some weaknesses that I have to indicate and would like authors to address:

1)  The final sample size is limited (n=10), which may have reduced statistical power and the ability to detect significant clinical effects.

2) Despite changes in microbiota diversity, no significant improvements in clinical outcomes were observed, raising questions about the clinical relevance of the findings. Maybe this method can be implemented in treatment of other diseases?
Why the specific changes you observed should influence UC specifically? Why do you view these changes as positive?

3)  All patients continued standard therapies, making it difficult to isolate the specific effects of Hâ‚‚ gas inhalation.

4) The absence of long-term follow-up data limits the assessment of safety and sustained efficacy.

I understand that these weaknesses cannot be avoided in an already existing dataset, but I suggest authors discuss these points in the Discussion section.

There are also specific comments through the text:

5) Table 1. Please check if the formatting is right and determine what H/P mean.

6)  Line 179: I suppose this should be deleted from text: “The following formatting styles are meant as a guide, as long as the full citation is complete and clear, Frontiers referencing style will be applied during typesetting.”

-Discussion:

7) The discussion could be strengthened by addressing potential mechanisms linking microbiota changes to clinical outcomes, even in the absence of significant effects.

8) Author Contributions: Some initials end in a period, some don't. Please unify them.

Decision: Major revisions.

Author Response

Reviewer 2

1, There are still some weaknesses that I have to indicate and would like authors to address:

 1)  The final sample size is limited (n=10), which may have reduced statistical power and the ability to detect significant clinical effects.

Response:

The study was originally designed to include 20 participants; however, an interim analysis of clinical efficacy was conducted when 10 participants had been enrolled. This analysis revealed that no clear efficacy had been observed in the clinical evaluation, and the study was consequently terminated after 10 participants had been enrolled. As previously mentioned, this results in a small-scale study; however, since the study targeted patients with disease activity, it was considered difficult to continue the intervention trial given the limited improvement in clinical symptoms.

In response to your suggestion, we have added a more detailed explanation of this in the Materials and Methods section (Lines 113-116), and the revised text has been highlighted in blue.

We also added a detailed description of this limitation The previous brief mention of this limitation has been removed to avoid redundancy in the last paragraph.

The revised content is as follows:

As the clinical evaluation did not show clear therapeutic effects, continuing the interventional trial without evident clinical improvement could not be justified from both scientific and ethical standpoints. Consequently, the study was terminated after enrolling 10 participants (Lines 115-118).

This study has several limitations. First, the sample size was small, as the trial was terminated after the interim analysis of 10 participants showed no clear clinical improvement. Although microbiota-related changes were observed, the limited sample size provided a reduced statistical power (Lines 315-318).

 2,  Despite changes in microbiota diversity, no significant improvements in clinical outcomes were observed, raising questions about the clinical relevance of the findings. Maybe this method can be implemented in treatment of other diseases?

 Why the specific changes you observed should influence UC specifically? Why do you view these changes as positive?

Response

A decrease in gut microbiota diversity has been noted in UC and is linked to disease severity. Restoring diversity can be considered a positive outcome, but it did not significantly alter the intestinal environment or lead to therapeutic effects. For example, combining hydrogen inhalation therapy with FMT may contribute to therapeutic effects. In other dysbiosis-related diseases, improving gut microbiota diversity is expected to aid treatment; however, whether hydrogen inhalation alone can achieve therapeutic effects, as in this study of UC, remains uncertain. We have added this content to the Discussion.

The revised content is as follows:

Recent studies have shown that a reduction in gut microbiota diversity is associated with UC severity. FMT introduces healthy donor microbiota into patients and can help restore microbial diversity, with greater increases in diversity following FMT being associated with better clinical outcomes [30]. In this study, although hydrogen gas inhalation did not lead to significant changes in the intestinal environment or clinical improvements, it successfully increased β-diversity variation of the gut microbiota compared to the placebo group (Lines 287-293).

This modest change may represent a small but meaningful step toward achieving the therapeutic potential of FMT in UC. Combining hydrogen gas inhalation with FMT may enhance therapeutic efficacy and warrants further investigation (Lines 302-305).

Therefore, the combination of FMT and H2 inhalation has the potential to alter and maintain intestinal microbiota (Lines 305-306)

 3, All patients continued standard therapies, making it difficult to isolate the specific effects of Hâ‚‚ gas inhalation.

Response:

We appreciate your insightful comment. Indeed, we agree that the ongoing treatment may have influenced the observed effects of H2 gas inhalation. Ideally, discontinuing existing treatments to isolate the specific impact of H2 gas would provide a more accurate assessment.

However, considering the potential risk of disease exacerbation in patients with ulcerative colitis, we determined that withdrawing ongoing treatments would be ethically inappropriate. Therefore, we adopted a design in which H2 gas inhalation was administered as an adjunct to the patients’ standard therapies.

We have now addressed this limitation in the Discussion section and acknowledged it as a potential confounding factor that should be considered in future studies.

The revised content is as follows:

Second, all patients continued standard therapies during the intervention, which may have confounded the results. Discontinuing these treatments to isolate Hâ‚‚ effects was deemed ethically inappropriate. Thus, Hâ‚‚ inhalation was administered as an adjunctive therapy (Lines 318-321).

 4, The absence of long-term follow-up data limits the assessment of safety and sustained efficacy.

 I understand that these weaknesses cannot be avoided in an already existing dataset, but I suggest authors discuss these points in the Discussion section.

Response:

We fully agree with you that the lack of long-term follow-up data limits the ability to evaluate both the sustained efficacy and long-term safety of hydrogen gas inhalation therapy.

As the therapeutic response to H2 gas inhalation was insufficient in some patients, additional or alternative treatments were subsequently introduced during the clinical course. As a result, conducting a consistent long-term follow-up under the same treatment conditions proved to be difficult. We have now added this point as a limitation in the Discussion section, as suggested.

The revised content is as follows:

Third, the lack of long-term follow-up limits assessment of sustained efficacy and safety. In some cases, additional treatments were introduced, preventing consistent long-term observation. (Lines 321-323)

 There are also specific comments through the text:

 5, Table 1. Please check if the formatting is right and determine what H/P mean.

Response:

We have adjusted the table format to ensure uniform row height. H1–H5 represent the participant IDs assigned to the hydrogen inhalation group, and P1–P5 represent the participant IDs assigned to the placebo group. We have added this explanation below Table 1.

The revised content is as follows:

H1–H5, participant IDs assigned to the hydrogen inhalation group; P1–P5, participant IDs assigned to the placebo group (Lines 121,122).

6,  Line 179: I suppose this should be deleted from text: “The following formatting styles are meant as a guide, as long as the full citation is complete and clear, Frontiers referencing style will be applied during typesetting.”

Response:

As you pointed out, the sentence “The following formatting styles are meant as a guide, as long as the full citation is complete and clear, Frontiers referencing style will be applied during typesetting.” is not appropriate for inclusion in the main text. We have now deleted this sentence from the manuscript accordingly.

 -Discussion:

 7, The discussion could be strengthened by addressing potential mechanisms linking microbiota changes to clinical outcomes, even in the absence of significant effects.

Response: We recognize that this is a very important discussion point, as you pointed out. In conjunction with your comment in 2), we have inserted the following text in the discussion section.

The revised content is as follows:

The revised content is as follows:

Recent studies have shown that a reduction in gut microbiota diversity is associated with UC severity. FMT introduces healthy donor microbiota into patients and can help restore microbial diversity, with greater increases in diversity following FMT being associated with better clinical outcomes [30]. In this study, although hydrogen gas inhalation did not lead to significant changes in the intestinal environment or clinical improvements, it successfully increased β-diversity variation of the gut microbiota compared to the placebo group (Lines 287-293).

This modest change may represent a small but meaningful step toward achieving the therapeutic potential of FMT in UC. Combining hydrogen gas inhalation with FMT may enhance therapeutic efficacy and warrants further investigation (Lines 302-305).

Therefore, the combination of FMT and H2 inhalation has the potential to alter and maintain intestinal microbiota (Lines 305-306)

8,  Author Contributions: Some initials end in a period, some don't. Please unify them.

Response:

We have corrected the contributions and have formatted it according to the journal guidelines (Line 341-346).

The revised content is as follows:

Author Contributions:  Conceptualization, T.M., D.I., and M.H.; methodology, D.I.; software, T.M., and D.I.; validation, K.N., M.H. and M.O.; formal analysis, D.I., and R.O.; investigation, M.K., M.O., H.I. and D.I.; resources, T.M., D.I. and R.K.; data curation, H.M., and W.S.; writing—original draft preparation, T.M.; writing—review and editing, D.I.; visualization, T.M.; supervision, A.N.; project administration, K.N.; funding acquisition, D.I. All authors have read and agreed to the published version of the manuscript.

Reviewer 3 Report

Comments and Suggestions for Authors

Dear Authors,

The manuscript explores a novel and timely topic. The randomised, double-blind, placebo-controlled design adds scientific rigour. However, due to the small sample size and some areas of vague explanation, revisions are necessary to strengthen clarity, methodological transparency, and scientific impact.

  1. Sample size and study termination justification
  • Lines 109–114: The justification for stopping the trial at 10 participants is weak. The authors mention an “interim analysis” and insufficient prior data but do not describe any formal stopping criteria or whether futility or ethical considerations were involved. Please clarify the statistical or ethical rationale for halting the study prematurely.
  • Suggestion: Add a power calculation (even post hoc) to better contextualise the limitations.

  1. Primary endpoints not clearly supported
  • Lines 189–195: The Mayo score and CAI improved numerically in both groups but without statistical significance. The only significant change was the sum of the MES (p = 0.02). This outcome seems selective and underpowered and may not reflect robust clinical relevance.
  • Suggestion: Provide effect sizes and confidence intervals. Consider discussing type I error risk due to multiple comparisons.

  1. Functional genomics section requires expansion
  • Lines 235–240: The KEGG orthology results are interesting, but are poorly contextualized biologically.
  • Suggestion: Highlight key metabolic or immunological pathways enriched or depleted post-Hâ‚‚ treatment. Were any of these linked to UC pathophysiology or anti-inflammatory effects?

  1. Discussion needs more depth
  • Lines 248–286: While the discussion raises several good points, it tends to overstate the significance of non-significant findings.
  • Lines 253–254: The term "favourable shifts" is used, although the α-diversity changes were not statistically significant (p = 0.19).
  • Suggestion: Temper conclusions and highlight the exploratory nature of these findings. Add a short paragraph discussing implications for future trial design (e.g., longer follow-up, microbiota resilience studies, or stratification by UC subtype).

  • Figure 2 (Page 7): The genus-level bar graphs are rich but hard to interpret without labels or statistical overlays.
  • Figure 4 (Page 9): It’s unclear if panel (b) includes a confidence ellipse. Please clarify the axes and grouping logic in the PCA plot.

  • Lines 303–314: The study is funded by MiZ Co., Ltd., the manufacturer of the intervention device. This must be more prominently disclosed in the discussion as a potential source of bias.

This pilot trial provides an intriguing exploratory dataset and highlights a potentially novel adjunctive strategy for UC. However, due to underpowering, limited generalisability, and overinterpretation of findings, the current manuscript requires major revision prior to consideration for publication.

Author Response

Reviewer 3

1,  Sample size and study termination justification

Lines 109–114: The justification for stopping the trial at 10 participants is weak. The authors mention an “interim analysis” and insufficient prior data but do not describe any formal stopping criteria or whether futility or ethical considerations were involved. Please clarify the statistical or ethical rationale for halting the study prematurely.

Suggestion: Add a power calculation (even post hoc) to better contextualise the limitations.

Response:

We apologize for our insufficient explanation. As stated in the manuscript, we initially planned to recruit 20 participants. However, an interim analysis was conducted after enrolling 10 participants, and the results showed no clear clinical improvement. Therefore, we decided to terminate further recruitment at that point. In response to your suggestion, we have added a more detailed explanation of this in the Materials and Methods section.

As you rightly pointed out, this study has a small sample size. However, since the participants were patients with active disease, we considered it ethically and practically difficult to continue an intervention trial in the absence of observable clinical improvement.

In response to your suggestion, we have added a detailed description of this limitation, which is highlighted in blue. The previous brief mention of this limitation has been removed to avoid redundancy in the last paragraph.

The revised content is as follows:

As the clinical evaluation did not show clear therapeutic effects, continuing the interventional trial without evident clinical improvement could not be justified from both scientific and ethical standpoints. Consequently, the study was terminated after enrolling 10 participants (Lines 115-118).

This study has several limitations. First, the sample size was small, as the trial was terminated after the interim analysis of 10 participants showed no clear clinical improvement. Although microbiota-related changes were observed, the limited sample size provided a reduced statistical power (Lines 315-318).

2, Primary endpoints not clearly supported

Lines 189–195: The Mayo score and CAI improved numerically in both groups but without statistical significance. The only significant change was the sum of the MES (p = 0.02). This outcome seems selective and underpowered and may not reflect robust clinical relevance.

Suggestion: Provide effect sizes and confidence intervals. Consider discussing type I error risk due to multiple comparisons.

Response:

As you rightly pointed out, the Mayo score and CAI showed numerical improvements without reaching statistical significance, and only the sum of the MES demonstrated a significant change (p = 0.02). We fully acknowledge that this may be a selective outcome, and due to the small sample size, the statistical power is limited.

In response to your suggestion, we have now calculated and added the effect size for the between-group difference in the change of MES, which was 1.73. The 95% confidence interval for this effect size is [0.15, 2.65], indicating a potentially moderate to large effect. However, the wide confidence interval reflects the small number of participants and associated uncertainty. These values have been added to the Results section for clarity.

We have also addressed the possibility of Type I error due to multiple comparisons in the revised Discussion section, and clarified that these results should be interpreted with caution and verified in larger-scale studies.

The revised content is as follows:

The between-group difference in the change of the sum of MES was statistically significant (p = 0.02). The effect size (Cohen’s d) was calculated as 1.73, with a 95% confidence interval of [0.15, 2.65], indicating a potentially moderate to large effect. However, the wide confidence interval reflects the small sample size and associated uncertainty (Lines 205-209).

Finally, the observed MES reduction should be interpreted cautiously due to small sample size and potential Type I error. Larger, adequately powered studies are necessary to confirm these preliminary findings (Lines 323-325).

3, Functional genomics section requires expansion

Lines 235–240: The KEGG orthology results are interesting, but are poorly contextualized biologically.

Suggestion: Highlight key metabolic or immunological pathways enriched or depleted post-Hâ‚‚ treatment. Were any of these linked to UC pathophysiology or anti-inflammatory effects?

Response:

Thank you very much for your insightful comment. We appreciate the suggestion to add biological context to the KEGG orthology (KO) findings.

In our analysis, while KEGG pathway enrichment was not clearly observed, several individual KOs showed notable changes after Hâ‚‚ inhalation.  Out of 115 KOs, 83 were significantly enriched in the Hâ‚‚ group, including K00242, K18120, K00135, and K14534. These 4 KOs, which are involved in butanoate metabolism, are associated with amino acid metabolism, mucosal repair, redox balance, and microbial stress responses, respectively. Therefore, the enrichment of these KOs may reflect an enhanced functional capacity of the gut microbiota, associated with maintaining intestinal homeostasis, butanoate-related metabolic activity, and inflammation regulation in UC. Moreover, these KO-level changes may reflect functional shifts in the gut microbiome potentially relevant to UC pathophysiology.

We have revised the Discussion section accordingly to incorporate this perspective.

The revised content is as follows:

While KEGG pathway enrichment was not clearly observed, several individual KOs showed changes following Hâ‚‚ inhalation, including K00242, K18120, K00135, and K14534. These KOs are involved in butanoate metabolism, and promote intestinal epithelial repair and barrier function, regulate immune responses, and influence the composition of the gut microbiota. While these KO-level changes do not form coherent pathways, they may reflect modest functional shifts in the microbiome associated with Hâ‚‚ exposure (Line 296-302).

4, Discussion needs more depth

Lines 248–286: While the discussion raises several good points, it tends to overstate the significance of non-significant findings.

Lines 253–254: The term "favourable shifts" is used, although the α-diversity changes were not statistically significant (p = 0.19).

Suggestion: Temper conclusions and highlight the exploratory nature of these findings. Add a short paragraph discussing implications for future trial design (e.g., longer follow-up, microbiota resilience studies, or stratification by UC subtype).

Response:

We agree with the reviewer’s point that some statements in the Discussion may have overstated the interpretation of non-significant findings. Specifically, while the beta diversity showed improvement, the alpha diversity did not reach statistical significance (p = 0.19). In response to this, we have removed the phrase “favourable shifts in bacterial composition” from the manuscript to better reflect the exploratory nature of our findings.

Furthermore, as suggested, we have added a short paragraph in the Discussion to address considerations for future trial design. In particular, we have noted the potential value of combining hydrogen gas inhalation with microbiota-directed therapies (e.g., FMT or probiotics), as well as the importance of longer-term follow-up and possible stratification by UC subtypes to better assess microbiota resilience and clinical relevance. We appreciate the reviewer’s comments, which helped us refine the interpretation and scope of our discussion.

The revised content is as follows:

Moreover, our study revealed some trends in microbial composition, such as improvement in beta diversity, but these findings should be interpreted with caution due to the lack of statistical significance in alpha diversity and the small sample size. Collectively, these observations highlight the exploratory nature of this study and suggest directions for future research (Lines 268-272).

To strengthen the clinical relevance of hydrogen gas inhalation therapy, future trials should consider longer-term follow-up to assess the durability of microbial shifts and clinical responses. In addition, combining hydrogen gas therapy with microbiota-targeted interventions such as FMT or probiotics may provide synergistic effects. Stratification by UC subtypes or disease activity may also help to clarify which patient populations are most likely to benefit from such therapies (Lines 308-313).

5, Figure 2 (Page 7): The genus-level bar graphs are rich but hard to interpret without labels or statistical overlays.

Figure 4 (Page 9): It’s unclear if panel (b) includes a confidence ellipse. Please clarify the axes and grouping logic in the PCA plot.

Response:

Figure 2 was intended to visually illustrate genus-level changes in microbial composition before and after H2 gas inhalation, rather than to present statistically significant differences. Given the small sample size in this study, it was challenging to apply reliable statistical comparisons at the genus level. Therefore, we aimed to provide an overview of the relative abundance of the top 20 genera to highlight potential compositional shifts in an exploratory manner.

Figure 4 Panel (b) shows a Principal Coordinates Analysis (PCoA) based on Bray-Curtis dissimilarity, where PC1 and PC2 explain 20.5% and 13.3% of the variation, respectively. Each arrow represents the temporal shift of an individual from baseline (circle) to 8 weeks (triangle). The colour indicates group assignment (blue for Hâ‚‚, grey for placebo).

Due to the small sample size (n = 5 per group), confidence ellipses were not applied, as they may not be statistically reliable under these conditions. Instead, we chose to display individual trajectories to emphasize within-subject variation over time. We have explained this in the legend and updated Figure 4b.

The revised content is as follows:

Figure 4.

(b) Principal coordinates analysis (PCoA) based on Bray-Curtis dissimilarity was performed to visualize β-diversity. PC1 and PC2 explain 20.5% and 13.3% of the variance, respectively. Each arrow represents the temporal shift of an individual subject from baseline (circle) to 8 weeks (triangle), with colour indicating group (blue: Hâ‚‚, grey: placebo). Due to the small sample size (n = 5 per group), confidence ellipses were not included, as they may not be statistically meaningful under these conditions (Lines 244-248).

6, Lines 303–314: The study is funded by MiZ Co., Ltd., the manufacturer of the intervention device. This must be more prominently disclosed in the discussion as a potential source of bias.

Response:

Thank you for pointing out this important matter.

We acknowledge that the study was funded by MiZ Co., Ltd., the manufacturer of the hydrogen gas inhalation device. At the outset of the study, we had a contractual agreement with the company that the results would be published regardless of whether they were clinically positive or negative. In the present study, as the clinical outcomes did not demonstrate significant improvement, we believe that the funding did not introduce a favourable bias. However, if you believe this information should be included in the discussion, we are willing to comply.

Round 2

Reviewer 1 Report

Comments and Suggestions for Authors

Good revision, and I recommend for publication. Congratulations!

Reviewer 2 Report

Comments and Suggestions for Authors

All comments were addressed. Paper can be accepted.

Reviewer 3 Report

Comments and Suggestions for Authors

My comments have been addressed. No further changes are required